# Inorganic Polymeric Materials for Injured Tissue Repair: Biocatalytic Formation and Exploitation

**DOI:** 10.3390/biomedicines10030658

**Published:** 2022-03-11

**Authors:** Heinz C. Schröder, Xiaohong Wang, Meik Neufurth, Shunfeng Wang, Rongwei Tan, Werner E. G. Müller

**Affiliations:** 1ERC Advanced Investigator Group, Institute for Physiological Chemistry, University Medical Center, Johannes Gutenberg University, Duesbergweg 6, 55128 Mainz, Germany; hschroed@uni-mainz.de (H.C.S.); wang013@uni-mainz.de (X.W.); mneufurt@uni-mainz.de (M.N.); shunwang@uni-mainz.de (S.W.); 2Shenzhen Lando Biomaterials Co., Ltd., Building B3, Unit 2B-C, China Merchants Guangming Science Park, Guangming District, Shenzhen 518107, China; tanrw@landobiom.com

**Keywords:** biomaterial, biosilica, polyphosphate, silicatein, alkaline phosphatase, nanoparticle, coacervate, morphogenetic activity, energy delivery, tissue regeneration

## Abstract

Two biocatalytically produced inorganic biomaterials show great potential for use in regenerative medicine but also other medical applications: bio-silica and bio-polyphosphate (bio-polyP or polyP). Biosilica is synthesized by a group of enzymes called silicateins, which mediate the formation of amorphous hydrated silica from monomeric precursors. The polymeric silicic acid formed by these enzymes, which have been cloned from various siliceous sponge species, then undergoes a maturation process to form a solid biosilica material. The second biomaterial, polyP, has the extraordinary property that it not only has morphogenetic activity similar to biosilica, i.e., can induce cell differentiation through specific gene expression, but also provides metabolic energy through enzymatic cleavage of its high-energy phosphoanhydride bonds. This reaction is catalyzed by alkaline phosphatase, a ubiquitous enzyme that, in combination with adenylate kinase, forms adenosine triphosphate (ATP) from polyP. This article attempts to highlight the biomedical importance of the inorganic polymeric materials biosilica and polyP as well as the enzymes silicatein and alkaline phosphatase, which are involved in their metabolism or mediate their biological activity.

## 1. Introduction

The oxides of two physiological, non-carbon elements have aroused particular interest in manufacturing tissue engineering scaffolds: silicon and phosphorus. The oxides of these elements and their polymers, silica and polyphosphate (polyP), are involved in the formation of the biomineral skeletons of a large number of animal species, from sponges (silica) to vertebrates (calcium phosphate and hydroxyapatite). While phosphorus is an essential element of life [1] and, as phosphate ion or phosphate group, has a multitude of functions in all organisms, including energy transfer [2,3], the function of silicon and silica, as a skeletal element, is mainly limited to siliceous sponges [4] and diatoms [5], in addition to plant phytoliths or silica bodies [6]. Nevertheless, silicon has been recognized as an essential trace element in many organisms including vertebrates, particularly for their bone formation [7,8].

Today it is widely accepted that biological mineralization or biomineralization processes, in contrast to purely chemical mineralization, begin with the formation of amorphous minerals. Crystalline biominerals such as bone hydroxyapatite are formed from the amorphous precursors [9]. This is important with regards to the potential use of (bio)minerals for the regeneration of bone tissue, since only the amorphous minerals such as amorphous silica, amorphous calcium carbonate (ACC) and amorphous calcium phosphate (ACP), but not the crystalline minerals show morphogenetic, osteoblastic activity, both in vitro and in vivo (for reviews see [10,11]).

Furthermore, it has become increasingly clear that, in animals, biomineral formation is largely enzyme-catalyzed. This has been shown in recent years for the principle classes of biominerals, for silica, calcium carbonate and calcium phosphate/hydroxyapatite. Amorphous silica (“biosilica”) is synthesized by the enzyme silicatein (formation of the silica skeleton of the siliceous sponges; [12]), ACC by the enzyme carbonic anhydrase [10] and ACP by the enzyme alkaline phosphatase (ALP) [3]. From these amorphous biominerals the crystalline minerals arise, such as calcite from ACC (formation of the calcite skeleton of the calcareous sponges) or hydroxyapatite (HA) in bones from ACP. In the latter case, it has even been shown that two biomineralization processes work closely together. The formation of HA is initiated by the deposition of biocatalytically formed ACC (carbonic anhydrase-mediated), which is then converted into ACP (involvement of ALP) and finally into crystalline HA (non-enzymatic) [11]. The phosphate used for the formation of ACP from ACC is supplied by polyP, as well as β-glycerophosphate. In contrast, biomineral deposition in non-animal organisms appears to be largely non-enzymatic, such as in diatoms and plants [5,6].

Besides the enzymes that synthesize the polymeric material, such as silicatein, which forms silica from monomeric precursors, the enzymes that degrade these materials are likewise important. Some of these enzymes, such as the bacterial polyP kinases, show both polymerizing and depolymerizing activity [13] or are involved in regeneration of metabolic energy needed for tissue function [3]. Due to their prominent role in the formation and function of materials that are of interest in biomineral-based approaches in regenerative medicine, the two basic enzymes present in animals, silicatein and ALP, as well as their application in tissue repair are discussed in more detail.

## 2. Differential Characteristics of the Si-O-Si and P-O-P Linkages

The properties of the Si-O-Si bond in silica, formed by silicatein, and of the P-O-P bond in polyP, cleaved by ALP, are quite different. While polyP is a stable polymer in aqueous solution in the neutral pH range, polysilicic acids are only stable at alkaline pH values [14]. The phosphoanhydride (P-O-P) bond in polyP, but not the siloxane (Si-O-Si) bond in silica, can be used for metabolic energy production. Due to its kinetic stability at physiological pH, polyP can act both as a storage and as a donor of metabolic energy, the release of which requires an enzyme, which lowers the activation energy (the ALP) or a decrease in pH (acidic pH) [3]. The differing behavior of the Si-O-Si and P-O-P bonds is reflected in the different activation energies (*E_a_*) for the hydrolysis of these bonds, which are much higher for P-O-P (*E_a_* > 104 kJ·mol^−1^, in neutral solution) [15] than for the hydrolysis of the Si-O-Si bond, which ranges from 50–70 kJ·mol^−1^ [16,17]. The *E_a_* of the acid-catalyzed hydrolysis of the P-O-P bond, on the other hand, is only 57 kJ/mol [15]. It is to be expected that an enzyme such as silicatein also lowers the activation energy of the silica polymerization reaction (calculated *E_a_* for the non-enzymatic Si-O-Si bond formation in neutral solution, 127 kJ·mol^−1^ [18]). The latter, non-enzymatic reaction proceeds via a transition state with the formation of a five-coordinated Si species and simultaneous proton transfer, resulting in the release of water [19]. The monomeric building blocks of polyP and of the polymeric silica also differ significantly in their p*K*_a_ values, which for orthophosphate are p*K*_a_ 2.2, 7.2 and 12.4 [20]; for orthosilicate the lowest p*K*_a_ is 9.5 [20].

## 3. Silicatein and Biosilica Formation

Silicatein is an enzyme that is exceptional for its unmatched property of catalyzing the synthesis of an inorganic material from inorganic monomeric precursors [12,21]. The existence of such an enzyme was new and unexpected. Silica, the product of the enzyme reaction, is an amorphous material that consists of a network of SiO_4_ tetrahedra connected to one another by Si-O-Si bonds with a varying range of angles. It differs from quartz, the crystalline form of silicon dioxide. Silica is a thermodynamically metastable material. It remains amorphous at temperatures below the glass transition temperature *T*_g_ = 1480 K, in contrast to the melt, which is obtained above the melting temperature *T*_m_ = 1986 K [22].

The silica synthesizing enzyme, silicatein, actually presents two enzyme activities: a hydrolase activity and a polymerase activity (Figure 1A). The hydrolase activity was first described by the group of Morse [23,24]. This activity of silicatein is responsible for the enzyme-catalyzed hydrolysis of tetraethyl orthosilicate (TEOS), a synthetic ester of orthosilicic acid. The enzymatic nature of the hydrolytic cleavage of the ester linkage was demonstrated by the finding that the reaction follows Michaelis-Menten kinetics; the Michaelis constant (*K_m_*), determined for bis(*p*-aminophenoxy)-dimethylsilane, is 22.7 μM [21]. It has been suggested that the monomeric orthosilicic acid product generated during this reaction begins to polymerize, leading to the formation of silica deposits [24].

Figure 1C shows a structural model of a silicatein molecule with the catalytic triad amino acids serine (Ser), histidine (His), and asparagine (Asn) forming the active site of the enzyme. The silicateins are related to the cathepsin family of proteases, which contain the three amino acids, cysteine (Cys), His and Asn, in their active center. In the silicateins, the Cys residue is replaced by a Ser residue [23]. In the immature silicatein protein, the catalytic site of the enzyme is covered by a propeptide sequence. Only after the cleavage of this sequence is this site accessible for the substrate and the catalytic reaction can proceed.

Initially, the immediate product of silicatein reaction was believed to be a solid material [24]. However, it was soon found that this product is made up of soluble silicic acid polymers or a water-rich gel-like silica material that has to undergo a maturation process to become a hard, solid material. This process leads to the formation of solid silica structures such as the spicules of the siliceous sponge skeletons and is described in more detail further below.

### 3.1. Mechanism of Silicatein Reaction

Silicatein catalyzes the biosilica condensation reaction of its natural substrate orthosilicic acid or its synthetic precursor TEOS even at concentrations of <1 mM [25], which is lower than those required for the polycondensation of orthosilicic acid to occur (>1 mM) [26,27]. Therefore, the polycondensation reaction at low precursor concentrations, ambient-temperature, and near-neutral pH conditions usually used in assays to determine silicatein activity cannot be explained by a mechanism only based on a catalytic hydrolysis reaction (hydrolysis of TEOS) as described by Morse [24]. To explain the silica polymerase activity of silicatein, we proposed an alternative model [28]. 

Modelling and docking studies suggested that the biocatalytic reaction of silicatein with its natural substrate, orthosilicate, leads to the formation of cyclic trisilicic acid species. The cyclic silicic acid trimer [trisiloxane; (SiO)_3_(OH)_6_] has a significantly higher reactivity than monomeric silicic acid and can start the further condensation reactions [28]. Similarly to the model proposed by the group of Morse [24,29], the first step in our model is a nucleophilic attack (S_N_2 type) of the OH group of the Ser residue of the catalytic center of the enzyme on the silicon atom of the silicic acid (Figure 1C,D). This reaction is facilitated by a hydrogen bridge formation between the Ser OH and the imidazole group of the His residue of the catalytic triad amino acids, Ser, His and Asn (Figure 1E). This creates a pentavalent intermediate that is covalently bound to the Ser residue of the enzyme (Figure 1F [left]). The leaving OH group of the substrate, which is bound to the catalytic triad Asn residue via a hydrogen bridge, is then released as a water molecule (Figure 1F, reaction 1). Following the same principle, an OH groups of the covalently bound silicic acid molecule then performs a nucleophilic attack at the silicon of a second silicic acid molecule, followed again by the transfer of a proton and the release of water (Figure 1F, reaction 2). By repeating this step, an enzyme-bound silicic acid trimer is formed, which, after cyclization, is cleaved from the enzyme by formation of a third siloxane bond (Figure 1F, reaction 3). The released reactive cyclic silicic acid trimer then initiates the further condensation reactions with free, soluble silicic acid molecules.

Silicatein is not only able to catalyze the synthesis of inorganic silicon oxides (silica) but also of organosiloxane compounds via trans-etherification reactions, involving the transfer of the silyl group from a silyl ether to the hydroxy group of an alcohol [30]. The *k_cat_*/*K_m_* values for the hydrolysis of 4-nitrophenyl silyl ethers were found in the range of 2–50 μM^−1^ min^−1^ [30].

In addition to the enzymatic mechanism, non-enzymatic processes might contribute to the metal oxide deposition on the silicatein protein. Presumably, both the basic amino acid (Lys and Arg) as well His residues of the silicatein propeptide can interact with the OH groups on the silica surface via hydrogen bridge formation and electrostatic, ionic charge interactions [31]. Such interactions might be involved in the deposition of non-silicon metal oxides mediated by silicatein molecules [32].

In the sponge spicules, the organic silicatein component is built into the inorganic silica material in the form of a central axial filament, which is surrounded by cylindrically arranged silica lamellae, as well as silicatein protein embedded within or between the partially or completely fused silica nanoparticles. The atomic structure of silicatein together with its silica product with a resolution of 2.4 Å has been reported [33].

### 3.2. Silicatein Assembly

The monomeric silicatein molecules have a strong tendency to oligomerize or polymerize to form long filaments. These filaments can only be disintegrated at high concentrations of chaotropic agents (urea). They are located in the center (axial canal) of the sponge spicules and are visible with light and electron microscopical methods. The self-assembly process of the silicatein can be monitored in dynamic light scattering experiments using the recombinant protein [31]. The recombinant silicatein is a monomer or small oligomer with a hydrodynamic (Stokes) radius of 4 nm at urea concentrations > 2.5 M. At lower urea concentrations, at 2 M urea, aggregates of about 20 silicatein molecules (size 22 nm) are formed. These aggregates grow in length quickly as the urea concentration is further decreased (<2 M).

The basic entity of the silicatein filaments is most likely made up of four silicatein molecules. This tetrameric structure, which silicatein assumes, offers a solution for the following problem. At low substrate concentrations, in the range of the *K_m_* value of silicatein (22.7 μM of silicic acid; [21]), which is below the concentration for polycondensation (1 mM), the chemical equilibrium between the silicic acid oligomers formed by silicatein and the monomers should be on the side of the monomeric silicic acid. Therefore, the cyclic trisilicic acid molecules should hydrolyze to the monomers according to the chemical equilibrium. Modeling studies, however, indicate that the catalytic pocket of each of the silicatein monomers, which make up the silicatein tetramer, forms a tunnel-like structure, pointing to the center of the tetramer so that the silicic acid substrate can enter the tunnel from the outer opening and the synthesized cyclic silicic acid trimer can exit into the space within the tetramer. There, the trimers can interact with the hydroxy groups of the Ser clusters [12] of the silicatein molecules that are exposed towards the center of the tetramer. As a result, the oligomeric product of the enzyme reaction, presumably an initially soft and water-rich material, accumulates in the center of the tetramer and is finally pushed out from there through the spaces between the silicatein molecules. The released silica then begins to harden to form a solid, water-insoluble material that envelops the silicatein molecules. As a result, a lattice-like (inorganic) silica matrix is formed, which is complementary to the (organic) silicatein protein lattice, as was observed in X-ray diffraction and electron microscopical studies of the spicules of the hexactinellid sponge *Monorhaphis chuni* [34].

The tendency of silicatein to aggregate and precipitate limits its availability for biosynthetic applications. Several methods have been developed to increase the solubility of the protein, for example the preparation of a silicatein fusion protein using a Strep-II tag instead of the histidine [35]. A soluble silicatein fusion protein has also been constructed using a soluble protein tag (ProS2) along with a carbohydrate-binding module that allowed to immobilize the protein on cellulose [36].

### 3.3. Biosynthesis and Processing of Silicatein

The cDNAs/genes that encode for different isoforms of silicatein have been isolated from various sponge species and characterized [23,32]. The sequence analyses revealed that the silicateins are related to the cathepsin family of proteins but differ from this group of proteases in the amino acids that form the catalytic site and are involved in the mechanism of action of these enzymes (see above). Two silicatein isoforms, silicatein-α and silicatein-β, which form an α_2_β_2_ tetramer, have been isolated, cloned and characterized from the marine demosponge *Suberites domuncula* [12].

Silicatein is synthesized as an inactive precursor (pre-prosilicatein) molecule, which is processed to the mature silicatein. The maturation of silicatein was demonstrated by expression of a gene, which encodes for a fusion protein of prosilicatein with the bacterial trigger factor chaperone protein. The proteolytic cleavage of the 87-kDa fusion protein expressed in *Escherichia coli* at the thrombin cleavage site within the spacer region into the trigger factor and the 35-kDa prosilicatein led to the immediate autocatalytic cleavage of the prosilicatein into the mature 23-kDa silicatein. This reaction was followed by the self-assembly of the mature silicatein into long insoluble filaments, a process that could be prevented by urea.

## 4. Alkaline Phosphatase and (Poly)Phosphate-Based Materials

In addition to the silicateins, there is only one other group of enzymes that form an inorganic polymer, the polyP kinases or polyP polymerases, which catalyze the formation of polyP. In contrast to silicatein, these enzymes use an organic compound (adenosine triphosphate, ATP) as the source for the inorganic building blocks (of phosphate) of the inorganic polymer, polyP. The energy-rich bound γ-phosphate of ATP is stepwise incorporated into the growing polyP chain with the formation of a similarly energy-rich acid anhydride bond. Therefore, the equilibrium is not on the side of the product (polyP) and the ΔG^0^ of the reaction is almost zero.

In contrast to the polyP kinases/polymerases in bacteria and yeasts, there is very little knowledge about the mechanism of polyP synthesis in animals and humans [37]. Both the mitochondria [38,39] and the acidocalcisomes [40] have been implicated in the synthesis of polyP, whereby the acidocalcisomes serve as storage organelles of the polymer [41]. While the mitochondrial polyP synthesis is driven by a gradient/potential on the inner mitochondrial membrane [38], the acidocalcisomal polyP synthesis is probably mediated by a vacuolar transporter chaperone (Vtc) complex, which in yeast is located in the membrane of the acidocalcisomes [40]. On the other hand, with regard to medical application, more detailed insights into the mechanism of polyP degradation have been gained in recent years. At the forefront of polyP degradation is the ALP (Figure 1B).

The ALP is a ubiquitous enzyme found in both prokaryotic and eukaryotic organisms and has been shown to be the major polyP-degrading enzyme in humans [3,42,43]. This enzyme catalyzes the dephosphorylation of a variety of phosphate-containing substrate molecules such as pyrophosphate (PP_i_), ATP, adenosine diphosphate (ADP) and adenosine monophosphate (AMP), β-glycerophosphate, and pyridoxal-phosphate, as well as synthetic substrates (*p*-nitrophenylphosphate), but also phosphorylated proteins such as osteopontin [44,45]. In addition, this enzyme can also exhibit phosphotransferase activity, in particular in a coupled chemical reaction [46]; Figure 1B.

Figure 1G shows the structure of the ALP monomer with its central β-sheet and flanking α-helices, which are similar between the human enzyme and bacterial (*E. coli*) enzymes [44]. The catalytic site with the active Ser residues and the two Zn^2+^-occupied metal ion sites are shown in Figure 1H,I. The third metal ion site, which is occupied by Mg^2+^, is not indicated. The negatively charged metaphosphate species (labelled in red in Figure 1H) is bound to the enzyme via the Zn^2+^ ions and the guanidinium group of an arginine (Arg) residue (Figure 1H,I).

The polyP molecules, the products of polyP polymerases and substrates of ALP, are mainly linear polymers made of up to several hundreds of orthophosphate (P_i_) residues, which are linked by energy-rich phosphoanhydride (P-O-P) bonds [3,37,47]. They are found in almost all cells and tissues in humans and also in the blood serum/extracellular space [46,47,48]. The size of the physiological polyP, in human blood, is in the range of 50 to 100 P_i_ units [48,49]. In addition to the linear polymers, cyclic oligo/polyphosphates exist, at least synthetically, which usually consist of a smaller number, usually 3, 4 or 6, of P_i_ residues [50]. Branched phosphates (ultraphosphates) have not yet been found in nature, but are available synthetically using phosphoramidite chemistry [51]. Surprisingly, some short-chain ultraphosphates show significant stability in aqueous solution. Recently, evidence has been provided that branched phosphates can also be cleaved enzymatically by ALP, leading to a linearization of the molecules [52].

In humans, the ALP family of proteins comprises four tissue-specific isoforms: tissue-nonspecific ALP, which occurs in bone, liver and kidney, placental ALP, germ cell ALP, and intestinal ALP. The tissue-nonspecific ALP, as well as the intestinal ALP, is an ectoenzyme that forms a dimer that is anchored to the outer cell membrane via a glycosyl-phosphatidylinositol (GPI) anchor [44,53]. Each catalytic site of the dimer contains two Zn^2+^ ions and one Mg^2+^ ion, which are involved in the catalytic mechanism [44]. The GPI anchor is cleaved by specific phospholipases such as the phosphatidylinositol-specific phospholipase C (PI-PLC), which leads to the release of the enzyme into the extracellular compartment [54]. Therefore, TNAP is the most common ALP isozyme found in the circulating human plasma [55], in addition to a smaller amount of intestinal ALP [56] and very few placental ALP [57].

### 4.1. Hydrolytic Cleavage of Polyphosphate

PolyP is hydrolytically cleaved to P_i_ by ALP using a processive mechanism, i.e., the enzyme remains bound to the substrate (polyP) after each catalytic cycle until the polyP chain is completely degraded [42]. This mechanism is also reflected in the affinity constants (*K_m_* values) of the polyP for the enzyme, which decrease with increasing chain length of the polymer, e.g., for the intestinal ALP from 218 µM (PP_i_) to 27 µM (polyP_4_), 3.9 µM (polyP_18_) and 0.5 µM (polyP_77_); at pH 7.5 [42]. Due to their lower *K_m_* (and higher *k_cat_*/*K_m_* ratios), long-chain polyP molecules are degraded faster than shorter polyP chains, consistent with the processive mode of action of the enzyme. The *k_cat_*/*K_m_* calculated from the kinetic constants for the intestinal enzyme at pH 7.5, based on P_i_, are 2.48 nM^−1^·s^−1^ (polyP_4_), 4.35 nM^−1^·s^−1^ (polyP_18_), and 5.4 nM^−1^·s^−1^ (polyP_77_) [42]. These values are extremely high and are close to the diffusion limit of 1 nM^−1^·s^−1^. Such high *k_cat_*/*K_m_* ratios are typically observed for enzymes showing a processive mechanism, e.g., prolyl 4-hydroylase, which catalyzes the hydroxylation of proline residues in procollagen [58]. For comparison, the *k_cat_*/*K_m_* ratio for *p*-nitrophenyl phosphate as an ALP substrate, calculated from the *K_m_* (0.4 mM) and *k_cat_* (42.55 s^−1^) of the intestinal ALP [59], is much lower, with 0.11 µM^−1^·s^−1^ (at pH 9.5), than the *k_cat_*/*K_m_* for polyP, e.g., polyP_77_ with 2.32 nM^−1^·s^−1^ (at pH 9.5) [42].

### 4.2. Phosphotransfer Reactions

In addition to the hydrolase activity, which causes the hydrolytic cleavage of polyP or the dephosphorylation of other substrate molecules, evidence has been presented that ALP also has phosphotransferase activity [3], as shown in Figure 1B. The energy-rich phosphate units of the polyP molecules can be transferred to adenine nucleotides (AMP and ADP) if the ALP acts in concert with an adenylate kinase (ADK) [46]. The latter enzyme catalyzes the interconversion of ADP to AMP and ATP. Thereby the chemical energy stored in the phosphoanhydride bonds of polyP is converted into metabolically useful energy. Based on inhibitor experiments, the ALP-catalyzed cleavage of polyP leads to the phosphorylation of AMP to ADP, which is subsequently phosphorylated to ATP via the ADK reaction.

Based on the current knowledge about the mode of action of phosphotransfer reactions catalyzed by ALP, a mechanism for the generation of ADP from polyP by ALP and the generation of ATP by the combined action of ALP and ADK has been proposed [3]. It is likely that a reactive phospho intermediate, a metaphosphate intermediate (Figure 1I), is involved in this reaction, which is bound to the catalytic site of the enzyme (Figure 1G,H) before reacting with the second substrate, with AMP, with formation of ADP (Figure 1B). ADP then serves as substrate for the ADK in the coupled ALP/ADK reaction [3], as shown in Figure 1B. A scheme of the postulated mechanism is shown in Figure 1J. This mechanism (“dissociative mechanism”) of the ALP-catalyzed phosphotransfer reaction, which is characterized by the formation of a transiently existing metaphosphate species (reaction 1; transferred to AMP in reaction 2), differs from the alternative (less likely and energetically less favorable) associative and concerted mechanisms, which involve a pentavalent transition state. The coupling of the ADP formation by ALP with the ADK reaction, which interconverts 2 ADP molecules into AMP and ATP, serving as a substrate in a variety of strongly exergonic phosphotransfer reactions, shifts the equilibrium of the ALP/ADK reactions towards the final ATP product [3]. In this way, the energy stored in polyP is transformed in metabolically useful energy in the form of ATP.

In this context, it could be relevant that it has been shown that the tissue-nonspecific ALP is also involved in thermogenesis [60]. This enzyme catalyzes the dephosphorylating of phosphocreatine in a futile cycle discovered in adipocytes, which consists of the phosphorylation and dephosphorylation of creatine [61]. In these cells, this cycle and the ALP are closely associated with the mitochondria, which are also a main organelle in polyP metabolism. Hence, it is possible that polyP is involved in a similar mechanism that dissipates chemical energy as heat. Due to the fact that energy dissipation through thermogenesis counteracts the development of obesity, polyP could have a beneficial effect on metabolic disorders associated with obesity such as diabetes. A regulatory function of the tissue-nonspecific ALP in mitochondrial respiration and ATP production has also been reported for bone and muscle progenitor cells [62].

### 4.3. Further Functions

In addition to its role in energy metabolism as a donor of energy-rich bound phosphate residues, various functions in humans have been identified or ascribed to polyP in recent years. Many of these functions are associated with the structure of polyP consisting of a multitude of phosphate residues that are linked via high-energy acid anhydride bonds. It has been suggested that polyP functions as a regulator in blood coagulation and fibrinolysis [63] as well as in the complement system [64]. Physiologically, polyP is released from the blood platelets after activation, especially at injured tissue sites. The platelets contain a high concentration of polyP (~1 mM) [11] due to their high content of acidocalcisomes (dense granules). Moreover, polyP has been shown to be involved mineralization of bone-forming cells and bone formation both in vitro and in vivo [65,66,67]. PolyP acts as a source of phosphate for HA formation [43,68]. It strongly enhances HA deposition of osteoblast-like SaOS-2 cells, accompanied by an increased expression of bone morphogenetic protein 2 (BMP-2) and other proteins involved in osteoblast differentiation and function, such as ALP. Of particular importance for a potential application in tissue regeneration, e.g., wound healing and bone repair, is the ability of polyP to induce cell migration and microvascularization [69].

## 5. Different Forms and Phases of Biosilica and Bioinorganic Polyphosphate

### 5.1. Silica

The primary product of the silicatein reaction consists of soluble silica polymers or a soft, gel-like silica material (Figure 2A) that undergoes an aging/hardening process to become a solid material. In nature, the gel-like consistency of the primary product of silicatein reaction is important, as this property makes the silica product moldable and adaptable, allowing the material to assume, e.g., the specific shape of the sponge spicules.

Different processes are involved in the maturation of the biosilica material.

#### 5.1.1. Gelation

The initial silicatein product first undergoes a polymerization-induced phase separation process. In the siliceous sponges, a mucin/nidogen-like protein has been identified, which induces this sol-gel transition, phase separation process [70].

#### 5.1.2. Syneresis

In the condensation reaction catalyzed by silicatein, a large amount of water is formed, in addition to the water of the surrounding aqueous solution/reaction mixture, which is entrapped in the silica network. This water is removed by syneresis, a process that is characterized by the expulsion of water and is associated with an increasing crosslinking of the biosilica network (Figure 2B). In parallel, a hardening of the silica material is observed. In living organisms such as sponges, the removal of water proceeds via aquaporin channels. Experimentally, syneresis can be induced by the addition of poly(ethylene glycol) (PEG) [71]. This process leads to a considerable shrinkage of the material [71]. During syneresis, proteins, including silicatein, can become entrapped in the hardened inorganic material, as shown in Figure 3A,B. This gives the material the advantageous properties of an inorganic-organic hybrid material, such as improved mechanical properties, as in the siliceous sponge spicules (Figure 3C).

#### 5.1.3. Biosintering

Finally, a sintering-like process, termed “biosintering”, is observed (Figure 2C). The incorporated silicatein proteins are assumed to facilitate the fusion of the silica nanospheres, a process that normally requires high temperatures [72]. Obviously, the silicatein molecules arranged on the surface of the nanospheres cause a strong reduction in the activation energy, which enables attachment/neck formation between the 50–200 nm sized spheres at ambient temperatures.

### 5.2. PolyP–PolyP Nano/Microparticles

PolyP molecules are strong polyanions at neutral pH that carry a negative charge on each P_i_ unit and even two on the terminal P_i_ groups. The dissociation constants of the internal and of the first terminal hydroxy groups are pK_1_ = 2.2, while the second OH at the terminal P_i_ has a pK_2_ = 7.2 [73]. The negative charges most likely contribute to the high kinetic stability of polyP in neutral or alkaline solutions [37,74]. Decreasing the pH (increasing the H^+^ concentration) strongly reduces the activation energy (*E*_a_) for hydrolytic cleavage of the phosphoanhydride (P-O-P) bond, allowing the reaction to take place [15,75].

In tissue engineering/repair, polyP can be administered either in soluble form (Figure 2D), e.g., as Na-polyP, or as nano- or microparticles (Figure 2E). Only the amorphous polyP nano/microparticles are morphogenetically active. They are also more biocompatible compared to crystalline phosphate particles, such as the crystalline HA or tricalcium phosphate, which are mostly used in bone tissue engineering and repair. The amorphous polyP nano/microparticles are bioinspired because they mimic the natural polyP deposits in the acidocalcisomes. They can be synthesized from the salts of polyP with various divalent cations [76,77,78]. Based on these particles, various materials for tissue engineering and repair have been developed. The polyP nano/microparticles can be applied either alone, incorporated into (poly(d,l-lactide-*co*-glycolide) (PLGA) microspheres, or embedded in a hydrogel forming polymer network or paste, or in a bio-ink suitable for 3D printing or even 3D cell printing applications.

#### 5.2.1. Calcium-PolyP

The Ca-polyP nano/microparticles (usually used in the in the range 80 and 200 nm; see Figure 3D,E) strongly stimulate the activity of bone forming cells. PolyP acts as an inducer of the expression of BMP-2 [65], as well as of Runt-related transcription factor 2 (RUNX2), a marker of osteoblastic differentiation. The polymer also potently enhances the expression and activity of ALP, the main enzyme of bone mineralization, in osteoblast-like SaOS-2 cells [65]. Moreover, polyP causes an increased expression of osteoprotegerin (OPG), a cytokine involved in the pathogenesis of osteoporosis [79], without affecting the level of RANKL (receptor activator of nuclear factor κB ligand) [67]. OPG acts as inhibitor of osteoclastogenesis by sequestering RANKL. As a result RANKL cannot bind to its receptor RANK, which leads to an impaired maturation and function of bone-degrading osteoclasts [67]. The beneficial effect of the amorphous Ca-polyP particles on healing of bone defects was confirmed in animal experiments [80].

#### 5.2.2. Magnesium-PolyP

In contrast to the Ca-polyP nano/microparticles, which enhance RUNX2 expression, amorphous Mg-polyP nano/microparticles induce the expression of the transcription factor SOX9, a marker for chondrogenic differentiation. Moreover, an upregulation of the expression of aggrecan, ALP and collagen types 2A1 and 3A1 is found [81]. Therefore, these particles are of particular interest for cartilage a tissue, which has only a low repair capacity. This avascular, bradytrophic tissue consists mainly of an extracellular matrix in which only a comparatively few cells are embedded, which makes this tissue, e.g. articular cartilage, susceptible to osteoarthrosis. Based on amorphous Mg-polyP nano/microparticles, new scaffold materials have been developed that resemble natural cartilage in their viscoelastic behavior. These matrices made of polyP and the likewise polyanionic glycosaminoglycan hyaluronic acid cross-linked via Mg^2+^ bridges proved to be regeneratively active and supported the infiltration by chondrocytes [81].

#### 5.2.3. Strontium-PolyP

The mineralization-inducing activity of amorphous Sr-polyP nano/microparticles was even more pronounced than that of Ca-polyP particles [77]. The Sr-polyP particles also upregulate the expression of ALP, but in contrast to the Ca-polyP particles show only a slight influence on the expression of the *SOST* gene in osteocytes [77]. Sclerostin, the product of the *SOST* gene, is a Wnt antagonist that inhibits the differentiation and mineralization of bone cells via the canonical Wnt/β-catenin signaling pathway [82]. Consequently, the amorphous Sr-polyP particles show a stronger effect on osteogenesis than the Ca-polyP particles. The results of in vitro experiments were confirmed in animal studies using the rat critical-size calvarial defect model; already after an implantation period of 12 weeks, almost complete restoration of the bone defect was found [77].

#### 5.2.4. Zinc-PolyP

In order to develop a formulation for the treatment of non-healing, chronic wounds, amorphous Zn-polyP microparticles were integrated in collagen-based wound mats [78]. In addition to the polyP/collagen mats, a wound gel consisting of polyP nanoparticles in an alginate-gelatin hydrogel matrix was developed [83]. The polyP component was integrated through ionic (Zn^2+^) cross-linking to alginate and periodate-oxidized alginate, bound to gelatin via Schiff base formation. This Zn-polyP-nanoparticle/coacervate containing hydrogel matrix significantly enhanced the growth of human epidermal keratinocytes and cell migration/attachment [83].

#### 5.2.5. Amorphous Ca-Phosphate Stabilized with PolyP

PolyP can also be used for stabilization of the amorphous state of ACP particles in order to maintain their morphogenetic activity [84]. These particles are composed so-called Posner’s clusters [Ca_9_(PO_4_)_6_], with a size of 0.7–1 nm and a Ca/P ratio of 1.5 [85], which assemble to 30–100 nm large aggregates that are finally transformed into the crystalline HA [(Ca_10_(PO_4_)_6_(OH)_2_; Ca/P ratio, 1.67] [86]. PolyP retards the transformation of the particles into the Ca-phosphate/HA crystals. For the stabilization of the amorphous particles, a polyP content of ≥10% polyP is required [84]. Particles prepared by co-precipitation of calcium and phosphate in the presence of lower concentrations of polyP are crystalline. The polyP-stabilized ACP shows strong osteoblastic and vasculogenic properties. In vitro experiments revealed that the particles markedly enhance the expression of ALP and collagen type 1, as well as mineralization in SaOS-2 cells. In vivo, in the calvarial bone defect model of rabbits, the polyP-stabilized ACP encapsulated in PLGA microspheres exhibited a pronounced osteoinductive activity, already after a six-week healing period, which was clearly superior compared to crystalline Ca-phosphate and β-tricalcium phosphate (β-TCP) used as a control [84]. The formation of new bone tissue was associated with increased vascularization and expression of vascular endothelial growth factor (VEGF).

#### 5.2.6. PolyP Coacervates

The amorphous polyP nano/microparticles are not the final, biologically active form of the polymer, but are activated upon contact with body fluids/proteins. Thereby the particles are transformed into a coacervate [87]. This coacervate phase of polyP (Figure 2F and Figure 3F) shows the characteristic morphogenetic activity of the polymer and is more biocompatible, but less stable/more rapidly degradable than the nano/microparticles, which are characterized by high stability due to their negative zeta potential, which prevents the particle aggregation [87]. A polyP coacervate is also obtained after addition of metal ions to soluble polyP at neutral pH and an approximately stoichiometric phosphate:metal ion ratio, while the amorphous polyP nanoparticles are formed from polyP under alkaline conditions in the presence of an excess of these ions [87]. The coacervate formation proceeds via a phase separation of a more viscous liquid phase, which contain the fraction of the longer polyP molecules, and a less viscous liquid phase with the fraction of the shorter polyP molecules [88,89]. The adaptable matrix thus formed offers optimal conditions for the ingrowth and embedding of cells and their differentiation/proliferation [87].

## 6. Properties of Amorphous Silica and PolyP as Regeneratively Active Inorganic Polymers

### 6.1. Morphogenetic Activity

Both amorphous inorganic biomaterials, biosilica and bio-polyP, are characterized by the property of being morphogenetically active. This property makes these materials particularly interesting for regenerative medicine. Both materials have the ability to induce the differentiation of stem cells by activating the expression of specific genes [10,67,90]. Biosilica has not only osteoconductive, but also osteoinductive activity [10]. It is able to induce the mineralization/HA formation of bone-forming cells such as human osteoblast-like SaOS-2 cells; it is also biocompatible and biodegradable [67]. In addition, biosilica is capable of increasing the expression ratio of OPG and RANKL [67]. A decrease in this expression ratio is causatively involved in the pathogenic mechanism of osteoporosis. It should be noted biosilica from diatoms has also been shown to exhibit pronounced osteogenic activity [91,92].

In addition to this morphogenetic activity, the release of monomeric orthosilicate from the amorphous polymeric silica could contribute to the stimulatory effect of silica on bone mineralization. The nucleation of HA is promoted at lower silicate concentrations, in the range 0.05–0.5 mM [93]. Moreover, it could be shown that ACC, the precursor of the likewise amorphous ACP and crystalline HA, is stabilized by silicate, which prevents crystallization of the amorphous mineral [94]. Based on modeling studies, it has also been reported that cyclic silicic acid trimeric motifs on the silica surface mimic the apatite nucleation site on bone sialoprotein (BSP) [95,96]. This Ca^2+^ binding site on the BSP surface consists of three (modified) acidic amino acids (phosphoserine and two glutamic acids [97]), which stereochemically fit to the silanol (SiOH) groups of the silicic acid trimer on the silica surface.

In the case of polyP, the morphogenetic effect can even be modulated by the nature of the cationic counterion of the polyanionic polyP. The direction of the differentiation of mesenchymal stem cells (MSC) can be steered either in the osteogenic or in the chondrogenic direction. Ca- and Sr-polyP nano/microparticles preferentially induce the regeneration of mineralized bone tissue [11,46,76,98]. Mg-polyP nano/microparticles, on the other hand, preferentially promote the regeneration of cartilage [81]. Such a differential gene expression is exceptional and makes the polyP-based nano/microparticles or coacervates, as well as scaffolds/hydrogels containing them, excellent materials for the regeneration/repair of bone and cartilage defects, but also for other applications such as wound healing [99]. The addition of cytokines/growth factors, as is the case with many other materials for tissue regeneration/repair, is not required [77,98].

### 6.2. Generation of Metabolic Energy

A unique property of polyP, in addition to its morphogenetic activity and not shown by silica, is its ability to generate metabolically useful energy. Similarly to glycogen as an organic energy storage, the inorganic polyP acts as a storage for metabolically available energy, but with two major differences. Firstly, polyP in particulate form needs much less space to store the same amount of energy as glycogen, and secondly it can be used directly to generate ATP via the ALP-ADK coupling, while ATP production from glycogen requires a series of multistep metabolic pathways (glycolysis, citric acid cycle and respiratory chain/F_1_F_0_-ATP synthase). No other polymer is able to store as much energy in such a small space as polyP. In particular, polyP acts as a donor of metabolic energy not only within the cell, but also extracellularly, in the extracellular matrix, which does not have any mitochondria as the main energy producers in human/animal organisms [46,100].

The standard Gibbs free energy (ΔG^0^) for the hydrolysis of P-O-P bond in the linear polyP is about −30.5 kJ·mol^−1^, similar to the ΔG^0^ for the hydrolysis of the α-β and β-γ phosphoanhydride bonds in ATP. Therefore, a large amount of energy can be released by complete hydrolysis of a polyP chain, e.g., in the case of polyP with the physiological chain length of about 40 P_i_ residues (polyP_n_ with *n* = 40, containing 39 P-O-P linkages), about 1189.5 kJ·mol^−1^ (=39 × −30.5 kJ·mol^−1^; Figure 4A). Since the cyclic triphosphate contains one more energy-rich bond than the linear triphosphate, complete hydrolysis of the cyclic molecule results in a 1.5-fold higher ΔG^0^ (−91.5 kJ·mol^−1^) than the linear polyP_3_ (−61 kJ·mol^−1^; Figure 4B). This energy released during the hydrolytic ring opening of cyclotriphosphate can be used for phosphorylation reactions in synthetic chemistry [101,102].

It is well established that tissue repair is a highly energy-consuming process and, in particular, tissues with only a few cells, such as cartilage, cannot cover their energy demand through mitochondrial ATP production. The second enzyme involved in ATP generation, ADK, is not only found intracellularly, but is also associated with the plasma membrane, similarly to ALP, where it could work in a coupled manner with ALP to produce ATP in the extracellular space (Figure 1B). This is important because many processes that take place in the extracellular matrix (ECM) consume energy, not only during tissue repair, but also during tissue remodeling, such as in human bone [103,104]. This energy, in the form of ATP or ADP, is needed for the synthesis and assembly of the macromolecules that build up the ECM (collagen, elastin, glycosaminoglycans and proteoglycans), but also for the activity of extracellular kinases [105,106] and extracellular chaperone(-like) proteins [46,107]. The functional significance of polyP as a donor for metabolic energy in the form of ATP can be demonstrated in inhibitor experiments. It has been shown that the increase in ATP level in the cell supernatants after exposure of cell cultures to polyP is suppressed by adding the ALP inhibitor levamisole. The administration of the ADK inhibitor P^1^,P^5^-di(adenosine-5′) pentaphosphate (AP_5_A) causes a drop of the extracellular ATP/ADP ratio, i.e., an increase in ADP, the product of the ALP reaction [46].

It has not yet known whether ALP can also split and liberate the energy present in cyclic trimetaphosphate in addition to the linear molecules in human cells, like the enzyme from *E. coli* [108]. An inorganic triphosphatase that hydrolyses the linear triphosphate has been reported to be present in mammalian cells [109].

## 7. Routes of Administration of Biosilica and Polyphosphate for Tissue Regeneration/Repair

Various methods–three-dimensional (3D) printing (Figure 5A), electrospinning (Figure 5B) and encapsulation into microspheres (Figure 5C), have been used to administer and to exploit the unique properties of amorphous silica and amorphous polyP nano/microparticles to be morphogenetically active and, in the case of polyP, also to provide the metabolic energy required for tissue regeneration and repair.

### 7.1. 3D Printing

The 3D printing technology for the fabrication of tailor-made implants is developing at a rapid pace. In addition, there is a growing interest in 3D printing of cell-laden hydrogel scaffolds (3D bio-printing). The latter technique has proven difficult because the cells must be present in a sufficiently high concentration, which requires a sufficient supply of metabolic energy for cell proliferation/differentiation and maintenance of cell functions.

In a first approach, we developed a biosilica-alginate hydrogel that was 3D printable [90], as shown in Figure 5D. Osteoblast-like SaOS-2 cells embedded into the hydrogel showed an increased expression of the genes for BMP-2 and OPG, as well as an enhanced HA formation.

Later, polyP-based hydrogel materials were developed that were also suitable for 3D printing (Figure 5G). In particular, the property of polyP to act as a donor of metabolic energy, in addition to its morphogenetic activity, turned out to be particular useful for the development of bio-inks, supplemented with living cells, which for the first time enabled efficient growth and survival of the cells in the 3D printed construct. The first polyP-containing 3D-printable bio-ink was developed by embedding amorphous Ca-polyP particles into a poly(ϵ-caprolactone) (PCL) matrix. The tissue-like scaffolds fabricated with this bio-ink supported the ingrowth of SaOS-2 cells; the stimulatory effect on cell migration was paralleled with an increased expression of the cell attracting chemokine SDF-1α (stromal cell-derived factor-1α) [110]. Another bio-ink was based on Na-polyP, gelatin and alginate, complexed with Ca^2+^ ions. SaOS-2 cells embedded in this ink remain alive and proliferatively active after printing [111]. Recently, a bio-ink based on *N*,*O*-carboxymethyl chitosan/alginate/gelatin was developed, which was enriched with both soluble polyP (Na-polyP) and Ca-polyP nanoparticles (acting as a depot form); this bio-ink enabled the successful 3D bio-printing of mesenchymal stem cells (MSC) (Figure 5G). The 3D-printed metabolic energy-supplying matrix promoted the migration, growth and differentiation propensity of MSC to functionally active, mineralizing osteoblasts [112].

### 7.2. Electrospinning

A method older than 3D printing but also suitable for the production of advanced scaffold materials is electrospinning. We have shown that electrospun PCL nanofiber mats can be loaded with biosilica formed by *s*ilicatein [113], as shown in Figure 5E. The biosilica-containing mats were fabricated in a two-step process. First, a PCL solution supplemented with TEOS was used for electrospinning. Subsequently, the electrospun nanofiber network was incubated with silicatein. The biosilica cover on the electrospun fiber mats formed by the silicatein reaction was found to be morphogenetically active. It supported the growth and mineralization of SaOS-2 cells in vitro [113]. Another approach exploited the synergistic effect of amorphous Ca-polyP nanoparticles and retinol on cell growth compared to either of the components alone. Electrospun fiber mats consisting of poly(d,l-lactide) (PLA) were developed, which contained Ca-polyP nanoparticles with incorporated retinol (Figure 5H) [114]. MC3T3-E1 cells grown onto these mats showed increased expression of the genes encoding for leptin and the leptin receptor, as well as for fatty acid binding protein 4 (FABP4). Recently we have shown that Ca-polyP/ACC particles [80] integrated in poly(methyl methacrylate) (PMMA) can also be applied as surface coatings using the blow spinning technology.

### 7.3. Microspheres

Silica and silicatein, as well as polyP and Ca-polyP nano/microparticles, can also be administered directly after encapsulation in PLGA. In animal experiments (rabbits), the implantation of PLGA microspheres containing silica and silicatein (Figure 5F) resulted in enhanced bone regeneration compared to β-TCP controls [115]. An emulsion-based technique was also used to embed Ca-polyP nano/microparticles or Na-polyP [116] (Figure 5I) and polyP-stabilized ACC [80] in PLGA. The encapsulation has the advantage that it protects the polyP against hydrolytic attack by the ubiquitous ALP. After implanting the nano/microparticle-containing spheres into cranial defects of rats, the particles were disintegrated within 4 weeks. In parallel, cells were seen to invade the space initially occupied by the particles, which synthesized new bone mineral. The mechanical properties of the newly formed bone were already close to intact trabecular bone tissue after 12-weeks of implantation, in contrast to β-TCP as a control, which was less than 50% efficient [80].

## 8. Biohybrid Formation with Hydrogel Forming Polymers

In order to combine the morphogenetic and, in the case of polyP, energy-supplying function of the inorganic materials, silica and polyP, or polyP nano/microparticles, with the advantageous (mechanical, water-binding, hardenability, etc.) properties of organic polymeric materials, a number of hybrid materials has been prepared.

Silicatein and its substrate orthosilicate have been integrated into a Na-alginate-based hydrogel. The biosilica formed by the enzyme resulted in an increased growth and HA formation of SaOS-2 cells embedded into this matrix compared to cells encapsulated into the hydrogel without these additives or with orthosilicate alone [90]. In addition, an increased expression of the genes encoding BMP-2, carbonic anhydrase and collagen type 1 was found. These results show that the enzymatically formed biosilica gives the matrix morphogenetic activity that makes it suitable for 3D cell printing. An increased expression of BMP-2 and OPG, but not of RANKL, was also observed in SaOS-2 cells grown onto 3D printed scaffolds that had been biologized by impregnation with silica. Furthermore, it was found that the functionalization of a chitosan-graft-polycaprolactone matrix with biosilica by surface immobilization of silicatein confers osteogenic activity towards SaOS-2 cells, which resulted in increased cell viability, ALP activity and mineralization [117]. Moreover, the co-incubation of beads containing osteoblast-like SaOS-2 cells with beads containing osteoclast-like RAW 264.7 led to an enhanced expression of the gene encoding for OPG in SaOS-2 cells when the beads were based on a silica-containing Na-alginate hydrogel matrix. On the other hand, a reduced expression of the gene for the tartrate-resistant acid phosphatase in RAW 264.7 cells was found, indicating that under these conditions the differentiation of these cells is impaired.

By using polyP together with other polyanionic (but organic) polymers, materials were obtained that could be cured in the presence of divalent Ca^2+^ or Mg^2+^ ions. In this process, metal bridges are formed between the polymers. With this approach, it has been possible for us, in recent years, to develop materials with adapted hardness and viscoelastic properties for potential application in bone and cartilage repair. Some materials turned out to be applicable in 3D printing/bio-printing procedures. Organic polymeric materials used with polyP included negatively charged polysaccharides such as *N*,*O*-carboxymethylchitosan (a carboxymethylated chitosan, a polysaccharide composed of β-(1→4)-linked d-glucosamine and *N*-acetyl-d-glucosamine)*,* alginate (composed of α-d-mannuronic acid and β-l-guluronic acid), hyaluronic acid (d-glucuronic acid and *N*-acetyl-d-glucosamine), chondroitin sulfate (a sulfated glycosaminoglycan composed of *N*-acetylgalactosamine and glucuronic acid), and karaya gum (a polysaccharide consisting galactose, rhamnose and galacturonic acid), as well as collagen and poly(vinyl alcohol) (PVA). The addition of calcium ions to a hydrogel matrix formed by polyP and the likewise polyanionic *N*,*O*-carboxymethylchitosan and alginate resulted in the formation of organized bundles of the polymers and the hardening of the material by crosslinking Ca^2+^ bridges [98]. A material with viscoelastic properties similar to cartilage was prepared by cross-linking polyP and hyaluronic acid via Mg^2+^ bridges [81]. Using a freeze-extraction technique, a macroporous scaffold was obtained from polyP, chondroitin sulfate and collagen. Addition of Ca^2+^ ions to the matrix resulted not only in Ca^2+^ crosslinks but also in the in situ formation of Ca-polyP nanoparticles, which were lined up on the collagen fibers [118]. In situ generation of Ca-polyP nanoparticles was also observed in a cryogel fabricated from polyP, karaya gum and PVA [119]. This matrix was obtained by Ca^2+^-mediated ionic gelation of the karaya gum and intermolecular cross-linking of PVA by freeze-thawing. After exposure to medium/serum the Ca-polyP nanoparticles within the cryogel were transformed into the biologically active coacervate, which supported cell invasion and cell growth of human MSC into the porous scaffold [119]. The prepared porous polyP/karaya gum/PVA cryogel had viscoelastic properties similar to cartilage and muscle [120,121]. Animal experiments (rat muscle) using microspheres of the material prepared by an emulsion technology revealed that the implants are already replaced by granulation tissue after an implantation period 2 to 4 weeks [119].

## 9. Outlook

The currently available data, collected in vitro and partly in vivo in animal experiments, show a great potential of biosilica and polyP-based materials in tissue regeneration and repair. The first steps towards translation into the clinic are currently going on.

Very promising results have recently been achieved in patients with chronic wounds. Wound healing, including healing of chronic wounds (wounds that do not heal within 3 months) such as diabetic foot ulcers, leg ulcers and pressure sores, is an energy (ATP) consuming process [122]. This energy demand can be covered by the energy delivering polyP. Animal experiments with normal mice and diabetic mice with delayed wound healing revealed that wound healing is significantly accelerated by topical administration of amorphous polyP nano/microparticles [99]. In order to apply polyP for treatment of chronic wounds in patients, we applied a newly developed technique based on the integration of the polyP microparticles into mechanically compressed collagen mats [78]. Both amorphous Zn-polyP particles [78] and amorphous Ca-polyP particles [123] were used. It was found that the incorporated polyP particles are transformed into the coacervate, the biologically active and biodegradable form of the polymer, after exposure to protein-containing body fluids/wound secretions. The treatment of chronic wounds in all patients examined so far with the collagen mats supplemented with Ca-polyP microparticles resulted in a strongly accelerated re-epithelialization rate and a complete healing of the wound within up to 9 weeks. In order to achieve a retard effect, soluble, immediately available water-soluble Na-polyP was administered together with the polyP microparticles with delayed activation.

In the field of surgery and dental surgery, the first clinical applications are in preparation. The intended applications will take advantage of the pronounced osteoinductive and osteoconductive activity of the developed polyP-based materials. These materials are able to meet the requirements to become an optimal material for regeneration/repair of bone and cartilage tissue. Such a material should offer the cells an optimal microenvironment in order to differentiate and proliferate in a spatially and temporally controlled manner [124]. It is expected that these materials support the different phases of repair, including the recruitment of mesenchymal stem cells and their differentiation into the osteoblast lineage, as well as angiogenesis, extracellular matrix synthesis, and mineralization. Physiologically, polyP and the growth factors required to induce these processes are released from the platelets, which are involved in the formation of the hematoma after bone injury [125]. As a component of a scaffold material, polyP has the ability to induce the necessary growth factors. In addition to factors that promote osteoblast differentiation such as BMP-2 or mineralization such as ALP, the factors induced by polyP also include VEGF, which supports angiogenesis. In addition to the metabolic energy delivered by the energy-rich polyP itself, the vascularization of the developing bone tissue is essential for a continuous supply of oxygen and nutrients.

The first clinical studies on the use of polyP in human bone repair have started [126]. These studies concern the safety and assessment of the osteoinductivity of amorphous Ca-polyP microparticles used for alveolar cleft repair.

There is also a multitude of potential medical applications for biosilica/silicatein, to stimulate new bone formation/regeneration, in surgery/dentistry in the form of implantable/injectable beads or in combination with other hydrogel/scaffold materials, as coatings or as a component of bio-inks for 3D printing of customized implants. Clinical studies with biosilica/silicatein have not yet started. The promising results of the animal experiments also suggest a clinical application of this biomaterial, possibly in combination with polyP.

## Figures and Tables

**Figure 1 biomedicines-10-00658-f001:**
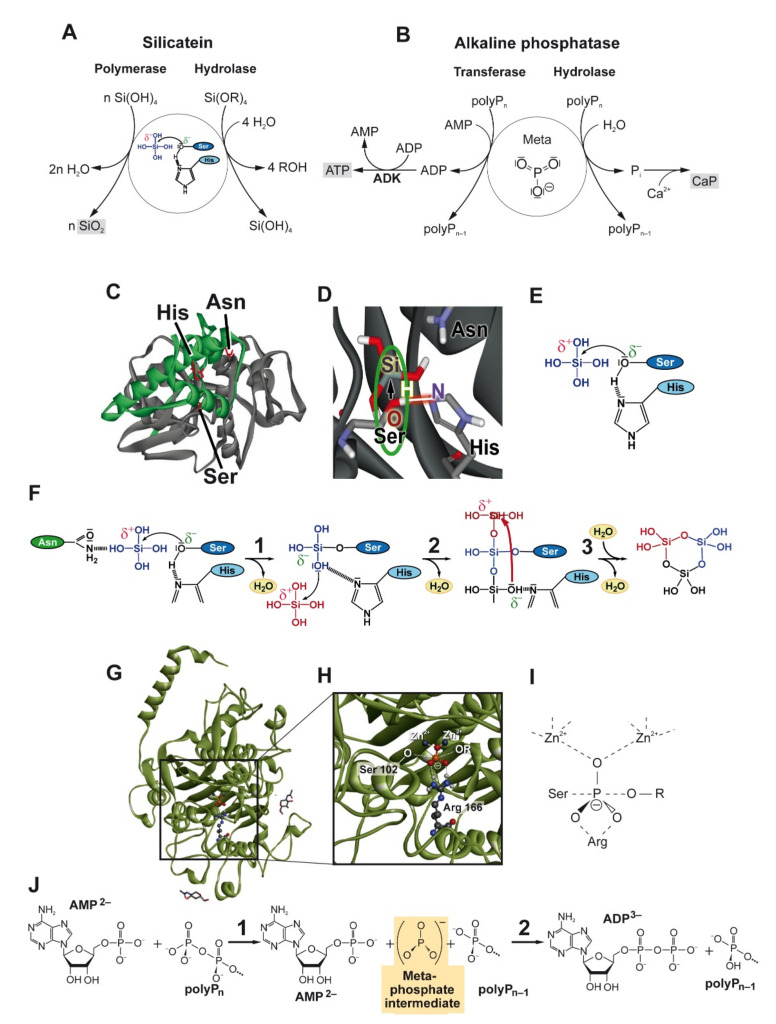
Biocatalytic activities of silicatein and alkaline phosphatase (ALP), the principle enzymes involved silica and calcium phosphate/hydroxyapatite biomineral synthesis. (**A**) Silicatein exhibits both a hydrolase activity (hydrolytic cleavage of Si-O-C bonds, e.g., of tetraalkoxysilane compounds), and silica polymerase activity, mediating the formation of polymeric biosilica from natural monomeric silicic acid precursors. (**B**) ALP can act both as a hydrolase, which degrades polyphosphate (polyP) to monomeric orthophosphate forming calcium phosphate deposits, and as a transferase that transfers a metaphosphate intermediate formed by cleavage of the polymer to AMP. The ADP formed is then used as a substrate by adenylate kinase (ADK). (**C**–**F**) Reactions proceeding at the catalytic site of silicatein and (**G**–**J**) of ALP; the reactions numbered are described in the text. (**C**) Computer model of silicatein showing the catalytic triad amino acids His, Ser and Asn (in red). The propeptide sequence of the immature silicatein is given in green. (**D**,**E**) Proposed mechanism of biosilica formation. The reaction starts by nucleophilic attack of the OH group of a Ser residue at the silicic acid monomer, facilitated by a hydrogen bridge formation to the His imidazole group. (**F**) Subsequent steps of the proposed silicatein mechanism, leading to the formation of a reactive cyclic trisilicic acid species. (**G**–**I**) Involvement of the zinc ions bound to the His imidazole rings of the catalytic center of ALP (**G**,**H**) in binding of the metaphosphate intermediate (**I**) during the enzymatic reaction. (**J**) Transfer reaction of the enzyme-bound metaphosphate species to AMP catalyzed by ALP.

**Figure 2 biomedicines-10-00658-f002:**
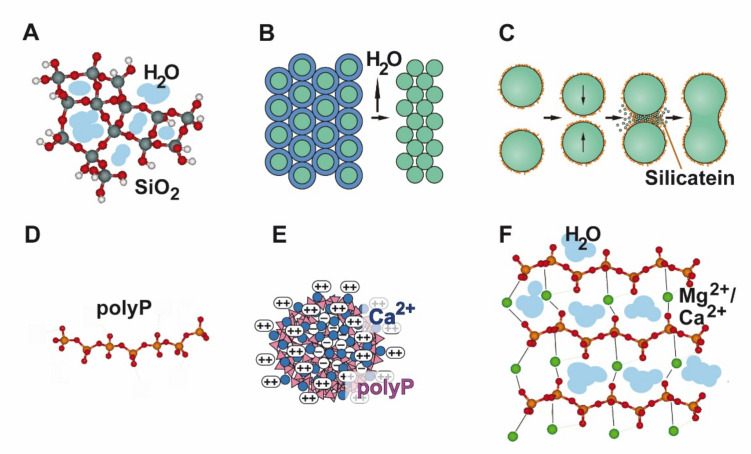
Different forms and phases of biosilica and bioinorganic polyP. (**A**–**C**) Enzymatically formed biosilica is obtained first as a water-rich gel-like product (**A**) that undergoes a hardening process by syneresis (**B**). In the presence of the silicatein, the formed amorphous spherical silica nanoparticles sinter together at ambient temperature under formation of solid biosilica blocks (**C**). (**D**–**F**) PolyP can be present either in a soluble form (**D**), e.g., as sodium salt (Na-polyP), or prepared in the form of nanoparticles (at alkaline pH in the presence of a stoichiometric surplus [based on phosphate] of divalent cations, e.g., Ca^2+^ (**E**) or as a water-rich coacervate (at neutral pH) (**F**).

**Figure 3 biomedicines-10-00658-f003:**
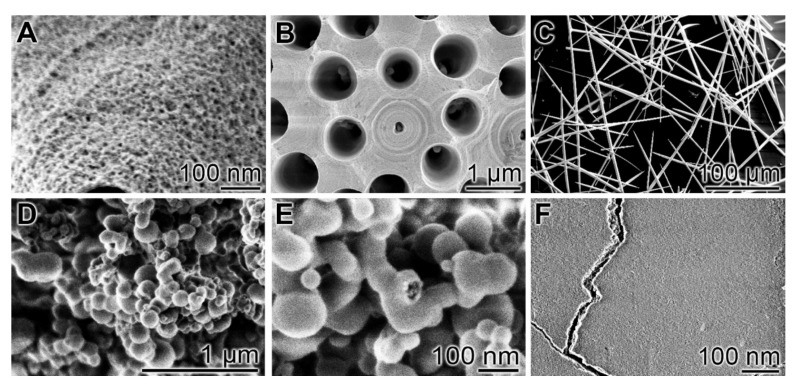
Biosilica and polyP based materials; scanning electron microscopy (SEM) analysis. (**A**–**C**): Biosilica. (**A**) Fusion of silica particles under (**B**,**C**) formation of solid silica structures. In (**B**), a sterraster of the demosponge *Geodia cydonium* and in (**C**) spicules of the demosponge *S. domuncula* are shown. The sterraster in (**B**) has been broken to make the silicatein axial filaments visible in the centers of the holes (axial canals). (**D**–**F**): PolyP. (**D**,**E**) Amorphous Ca-polyP nanoparticles (different magnifications. Adapted with permission from ref. [87]. Copyright 2018, John Wiley and Sons. (**F**) Coacervate formed by Ca-polyP.

**Figure 4 biomedicines-10-00658-f004:**
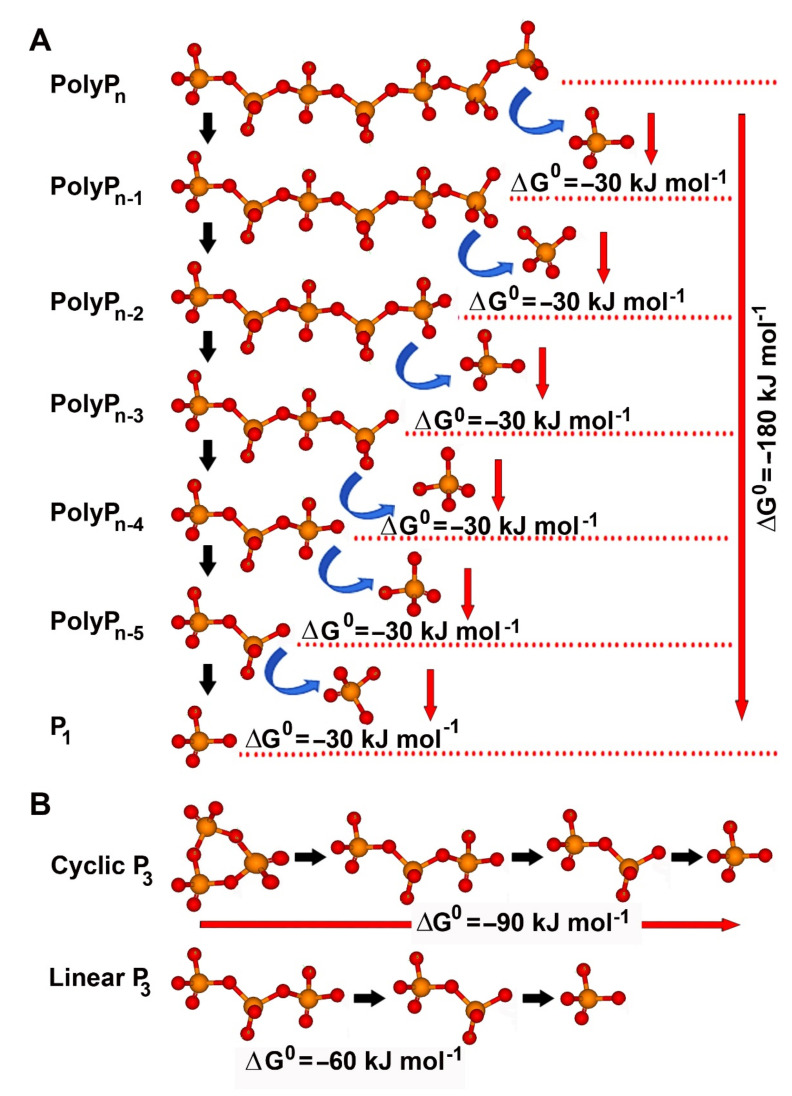
Delivery of metabolic energy during hydrolytic degradation of inorganic polyP. (**A**) Release of Gibb’s free energy (ΔG^0^) during stepwise hydrolysis of the energy-rich phosphoanhydride bonds of linear polyP molecules. During each hydrolytic step, a ΔG^0^ of approximately −30 kJ·mol^−1^ is liberated. (**B**) Short-chain triphosphate (polyP_3_) can exist either as a linear molecule or in a cyclic form. The cyclic polyP_3_ has one phosphoanhydride bond more than the linear molecule and can deliver 50% more energy (ΔG^0^ = −90 kJ·mol^−1^) compared to linear polyP_3_ (ΔG^0^ = −60 kJ·mol^−1^).

**Figure 5 biomedicines-10-00658-f005:**
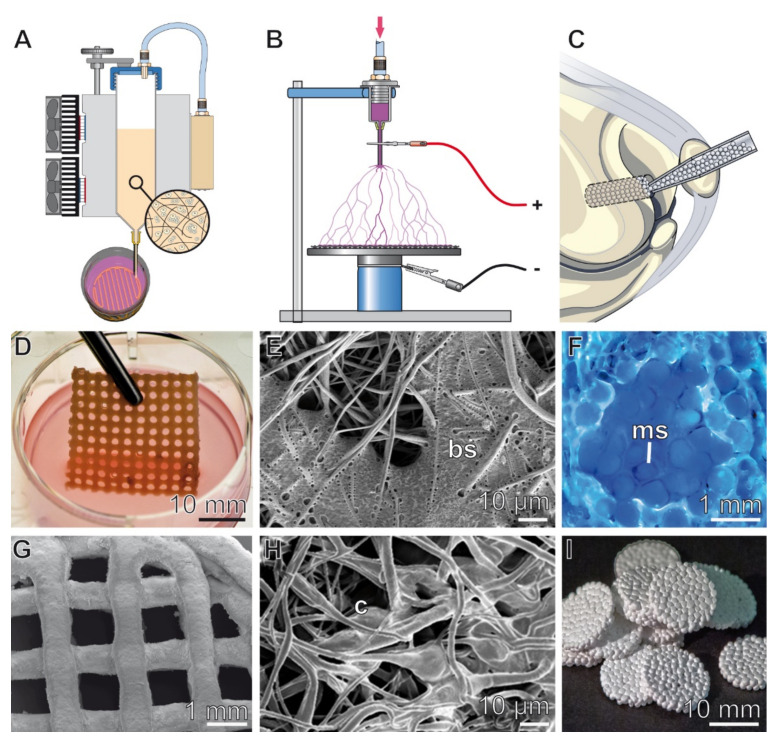
Different routes of administration of biosilica and polyP for tissue regeneration/repair. Both biosilica, either alone or together with silicatein, and polyP, as soluble polyP or polyP nanoparticles, can be applied by using (**A**) 3D printing techniques, (**B**) electrospinning, or (**C**) in a microparticular form, after encapsulation in poly(d,l-lactide-*co*-glycolide (PLGA). (**D**–**F**) Biosilica. (**D**) 3D printed grid of a biosilica-supplemented and SaOS-2 cells containing hydrogel. (**E**) Patches of biosilica (bs) deposits formed by silicatein on the surface of a TEOS-containing electrospun PCL nanofiber mat; SEM. Adapted with permission from ref. [113]. Copyright 2014, John Wiley and Sons. (**F**) Healing of a bone defect after implantation of PLGA-based microspheres, containing silica/silicatein. A tissue section from patella of rabbit in vivo labeled with oxytetracycline [115] is shown, allowing visualization of new bone formation; ms, microsphere. (**G**–**I**) PolyP. (**G**) 3D bio-printed disc prepared with a polyP-supplemented bio-ink solution, based on *N*,*O*-carboxymethyl chitosan; environmental scanning electron microscopy (ESEM). Adapted with permission from ref. [112]. Copyright 2022, IOP Publishing. (**H**) Electrospun fibrous mat fabricated with poly(lactic acid) (PLA) and amorphous Ca-polyP nanoparticles. After incubation with mouse calvaria MC3T3-E1 cells, the formation of cell (c) layers on the mats is visible. Adapted with permission from ref. [114]. Copyright 2015, Elsevier. (**I**) Discs prepared from amorphous polyP encapsulated into PLGA microspheres. Adapted with permission from ref. [116]. Copyright 2016, John Wiley and Sons.

## Data Availability

Not applicable.

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
