# Peer review of "Inorganic Polymeric Materials for Injured Tissue Repair: Biocatalytic Formation and Exploitation"

_biomedicines, 2022, doi:10.3390/biomedicines10030658_

Round 1

Reviewer 1 Report

Some conceptual mistakes or sentences should be re-organized or corrected:

  • Page 1, please revise line 35 “the function of silicon and sil- 35 ica, as an animal skeletal element, is mainly limited to sponges [4]”. Silicon and silica manipulation also belongs to diatom, spicule metabolism and some upper terrestrial plants.
  • Again the authors should revise the sentence in page 2, line 49 “Amor- 49 phous silica (“biosilica”) is synthesized by the enzyme silicatein (formation of the silica 50 skeleton of the siliceous sponges”, due to they refer only to sponges and not to diatoms and upper plants.
  • This referee does not like some sudden technical (too many) infos directly in the introduction parts, for example lines 65 and over (“The latter 65 activity, along with adenylate kinase (ADK), is involved in the synthesis of adenosine tri- 66 phosphate (ATP) - a key metabolite for successful tissue repair [3]. Similarly, silicatein has 67 both silica polymerase and hydrolase/esterase activity, as outlined below.”). The introduction should be more general, generalist and less technical.
  • Figure 1 is absolutely not readable. This referee cannot revise its content. Moreover, captions of the figures are too long.
  • Page 4, “Silicatein and Biosilica Formation”, line 113 and over: The explanation of the chemistry beyond the formation of biosilica is impressive and exhaustive for this referee. However, in the specific part dedicated to silicatein a full sentence about the physical states of these silica-precursors or adducts lack! For istance…are they nanoparticles, microparticles, organic-inorganic scaffolds? Are they fibers (the only physical state mentioned)…adding a material science point of view (few science) could be more complete! This referee read the specific parts dedicated to the physical states of these compounds and macromolecular aggregates, and he is always more worried about why the authors confidently separated the parts about the chemistry from the part about the physical states. This should strongly be tiring for a reader.
  • In the specific sections dedicated to silicatein and biosilica (1) and polyphosphate synthesis (2), a structural consideration (few sentences) about the enzyme domains lack!
  • The part about the manufacture techniques is exhaustive for this referee.
  • Some efficient literature about the use of biosilica for osteogenic purposes lacks: a. From polydisperse diatomaceous earth to biosilica with specific morphologies by glucose gradient/dialysis: A natural material for cell growth; Cicco, S.R.; b. In vivo functionalization of diatom biosilica with sodium alendronate as osteoactive material; Cicco, S.R.

Author Response

Referee: 1

Query:

  1. Some conceptual mistakes or sentences should be re-organized or corrected:

Page 1, please revise line 35 “the function of silicon and sil- 35 ica, as an animal skeletal element, is mainly limited to sponges [4]”. Silicon and silica manipulation also belongs to diatom, spicule metabolism and some upper terrestrial plants.

Answer:

Thanks dear referee – Now we included: “… siliceous sponges [4] and diatoms [5], in addition to plant phytoliths or silica bodies [6]. Nevertheless, silicon has been recognized as an essential trace element in many organisms including vertebrates, particularly ...”; with “5. Hildebrand, M; Lerch, S.J.L.; Shrestha, R.P. Understanding diatom cell wall silicification—Moving forward. Front. Mar. Sci. 2018, 5, 125.”; and “6. Guerriero, G.; Stokes, I.; Valle, N.; Hausman, J.F.; Exley, C. Visualising silicon in plants: Histochemistry, silica sculptures and elemental imaging. Cells 2020, 9, 1066.”.

Query:

  1. Again the authors should revise the sentence in page 2, line 49 “Amor- 49 phous silica (“biosilica”) is synthesized by the enzyme silicatein (formation of the silica 50 skeleton of the siliceous sponges”, due to they refer only to sponges and not to diatoms and upper plants.

Answer:

Thanks; we included: “In contrast, biomineral deposition in non-animal organisms appears to be largely non-enzymatic, such as in diatoms and plants [5,6].”.

Query:

  1. This referee does not like some sudden technical (too many) infos directly in the introduction parts, for example lines 65 and over (“The latter 65 activity, along with adenylate kinase (ADK), is involved in the synthesis of adenosine tri- 66 phosphate (ATP) - a key metabolite for successful tissue repair [3]. Similarly, silicatein has 67 both silica polymerase and hydrolase/esterase activity, as outlined below.”). The introduction should be more general, generalist and less technical.

Answer:

Thanks – you are right: We removed this information from the Introduction part, including the sentence “The latter activity, along with adenylate kinase (ADK), is involved in the synthesis of adenosine triphosphate (ATP) - a key metabolite for successful tissue repair [3]. Similarly, silicatein has both silica polymerase and hydrolase/esterase activity, as outlined below.”

Some parts of the Introduction were now included in a new, separate chapter: “2. Differential Characteristics of the Si-O-Si and P-O-P Linkages.”. The numbering of the following chapters therefore changed.

Query:

  1. Figure 1 is absolutely not readable. This referee cannot revise its content. Moreover, captions of the figures are too long.

Answer:

Thanks – please excuse tis mistake. We inserted Figure 1 with good resolution.

We tried to shorten the legends to the Figures as far as possible.

Query:

  1. Page 4, “Silicatein and Biosilica Formation”, line 113 and over: The explanation of the chemistry beyond the formation of biosilica is impressive and exhaustive for this referee. However, in the specific part dedicated to silicatein a full sentence about the physical states of these silica-precursors or adducts lack! For istance…are they nanoparticles, microparticles, organic-inorganic scaffolds? Are they fibers (the only physical state mentioned)…adding a material science point of view (few science) could be more complete! This referee read the specific parts dedicated to the physical states of these compounds and macromolecular aggregates, and he is always more worried about why the authors confidently separated the parts about the chemistry from the part about the physical states. This should strongly be tiring for a reader.

Answer:

You are correct. Therefore, we rearranged now the text in the Chapters 2 and 3 (now Chapters 3 and 4): “3. Silicatein and Biosilica Formation” and “4. Alkaline Phosphatase and (Poly)phosphate-based Materials”. To make these chapters better readable, we divided them in subchapters: “3.1. Mechanism of Silicatein Reaction”; “3.2. Silicatein Assembly”; “3.3. Biosynthesis and Processing of Silicatein”, “4.1. Hydrolytic Cleavage of Polyphosphate”; “4.2. Phosphotransfer Reactions”; and “4.3. Further Functions”.

In the specific part to silicatein, we  included a sentence about the physical state of the silica-precursor “Initially, the immediate product of silicatein reaction was believed to be a solid material [27]. However, it was soon found that this product is made up of soluble silicic acid polymers or a water-rich gel-like silica material that has to undergo a maturation process to become a hard, solid material. This process leading to the formation of solid silica structures like the spicules of the siliceous sponge skeletons is described in more detail further below.”

The detailed description of the maturation of the primary silicatein product to the mature, solid silica is described in Chapter 5. “Different Forms and Phases of Biosilica and Bioinorganic Polyphosphate”, “5.1. Silica - The primary product of the silicatein reaction consists of soluble silica polymers or a soft, gel-like silica material (Figure 2A) that undergoes an aging/hardening process to become a solid material. …”.

Query:

  1. In the specific sections dedicated to silicatein and biosilica (1) and polyphosphate synthesis (2), a structural consideration (few sentences) about the enzyme domains lack!

Answer:

Many thanks – now we added in Chapter “3. Silicatein and Biosilica Formation”: “Figure 1C shows a structural model of a silicatein molecule with the catalytic triad amino acids serine (Ser), histidine (His), and asparagine (Asn) forming the active site of the enzyme. The silicateins are related to the cathepsin family of proteases, which contain the three amino acids cysteine (Cys), His and Asn in their active center. In the silicateins, the Cys residue is replaced by a Ser residue [26,27]. In the immature silicatein protein, the catalytic site of the enzyme is covered by a propeptide sequence. Only after cleavage of this sequence is this site accessible for the substrate and the catalytic reaction can proceed.”;

and in Chapter “4. Alkaline Phosphatase and (Poly)phosphate-based Materials”: “Figure 1G shows the structure of the ALP monomer with its central β-sheet and flanking α-helices, which are similar between the human enzyme and bacterial (E. coli) enzymes [48]. The catalytic site with the active Ser residues and the two Zn2+-occupied metal ion sites are shown in Figure 1H and I. The third metal ion site, which is occupied by Mg2+, is not indicated. The negatively charged metaphosphate species (labelled in red in Figure 1H) is bound to the enzyme via the Zn2+ ions and the guanidinium group of an arginine (Arg) residue (Figure 1H and I).”.

Query:

  1. The part about the manufacture techniques is exhaustive for this referee..

Answer:

Thanks dear referee – we removed the sentences about the preparation methods of the nanoparticles in the Chapter “5.2. PolyP – PolyP Nano/microparticles”. Only the different biological effects of the particles are described.

Query:

  1. Some efficient literature about the use of biosilica for osteogenic purposes lacks: a. From polydisperse diatomaceous earth to biosilica with specific morphologies by glucose gradient/dialysis: A natural material for cell growth; Cicco, S.R.; b. In vivo functionalization of diatom biosilica with sodium alendronate as osteoactive material; Cicco, S.R.

Answer:

Many thanks for these new references. We added: “It should be noted biosilica from diatoms has also been shown to exhibit pronounced osteogenic activity [105,106].” and cited these papers: “105. Cicco, S.R.; Vona, D.; Leone, G.; Lo Presti, M; Palumbo, F.; Altamura, E.; Ragni, R.; Farinola, G.M. From polydisperse diatomaceous earth to biosilica with specific morphologies by glucose gradient/dialysis: a natural material for cell growth. MRS Communications 2017, 7, 214–220.” and: “106. Cicco, S.R.; Vona, D.; Leone, G.; De Giglio, E.; Bonifacio, M.A.; Cometa, S.; Fiore, S.; Palumbo, F.; Ragni, R.; Farinola, G.M. In vivo functionalization of diatom biosilica with sodium alendronate as osteoactive material. Mater. Sci. Engineer. C 2019, 104, 109897.”.

Reviewer 2 Report

This manuscript nicely reviewed various bioinorganic materials, including biosilica, bioinorganic polyphosphate, and other interesting materials. The work is well organized and "easy-to-read", but some improvements can be done:

1) Figure 1 is or poor quality, impossible to read the chemical equations. Improve the quality

2) Authors like a combination of capital and small letter for navigation inside the Figures. Sometimes it can be accepted but the authors overused this approach. I believe that sometimes you better label your Figs as 1a,b,c,d,e,f.... without mixing Aa,Ab,Ac..Ba, Bb....

Author Response

Referee: 2

Comments to the Author

  1. This manuscript nicely reviewed various bioinorganic materials, including biosilica, bioinorganic polyphosphate, and other interesting materials. The work is well organized and "easy-to-read", but some improvements can be done:

Answer:

Many thanks, dear referee, for this very positive assessment.

Query:

  1. 2) Authors like a combination of capital and small letter for navigation inside the Figures. Sometimes it can be accepted but the authors overused this approach. I believe that sometimes you better label your Figs as 1a,b,c,d,e,f.... without mixing Aa,Ab,Ac..Ba, Bb....

Answer:

Thanks dear referee – you are right. We removed the combination of capital and small letters in the Figures (this was the case in Figure 1 and Figure 2) and uses now Figure 1A, Figure 1B etc.; and Figure 2A, Figure 2B etc.